# Effect of Early Closing of the Inlet Valve on Fuel Consumption and Temperature in a Medium Speed Marine Diesel Engine Cylinder

**Vladimir Pelić [1]** , **Tomislav Mrakovčić [2],\*** , **Vedran Medica-Viola [2]** and **Marko Valčić [2]**

1   Faculty of Maritime Studies, University of Rijeka, Studentska ulica 2, 51000 Rijeka, Croatia; vpelic@pfri.hr
2   Faculty of Engineering, University of Rijeka, Vukovarska 58, 51000 Rijeka, Croatia;
    vmedica@riteh.hr (V.M.-V.); marko.valcic@riteh.hr (M.V.)
\*   Correspondence: tomislav.mrakovcic@riteh.hr; Tel.: +385-51-651-520

**Abstract:** The energy efficiency and environmental friendliness of medium-speed marine diesel engines are to be improved through the application of various measures and technologies. Special attention will be paid to the reduction in $NO_x$ in order to comply with the conditions of the MARPOL Convention, Annex VI. The reduction in $NO_x$ emissions will be achieved by the application of primary and secondary measures. The primary measures relate to the process in the engine, while the secondary measures are based on the reduction in $NO_x$ emissions through the after-treatment of exhaust gases. Some primary measures such as exhaust gas recirculation, adding water to the fuel or injecting water into the cylinder give good results in reducing $NO_x$ emissions, but generally lead to an increase in fuel consumption. In contrast to the aforementioned methods, the use of an earlier inlet valve closure, referred to in the literature as the Miller process, not only reduces $NO_x$ emissions, but also increases the efficiency of the engine in conjunction with appropriate turbochargers. A previously developed numerical model to simulate diesel engine operation is used to analyse the effects of the Miller process on engine performance. Although the numerical model cannot completely replace experimental research, it is an effective tool for verifying the influence of various input parameters on engine performance. In this paper, the effect of an earlier closing of the intake valve and an increase in inlet manifold pressure on fuel consumption, pressure and temperature in the engine cylinder under steady-state conditions is analysed. The results obtained with the numerical model show the justification for using the Miller processes to reduce $NO_x$ emissions and fuel consumption.

**Keywords:** marine diesel engine; Miller process; fuel consumption; nitrogen oxides

## 1. Introduction

Maritime transport is extremely important for the exchange of goods at the global level. Although the transport of goods by sea is the most efficient known mode of transport in terms of energy consumption per mile travelled and per tonne of cargo transported, it faces increasing demands in terms of energy efficiency and the reduction in negative environmental impacts. The requirements for reducing air pollution with pollutants from marine power plants are defined in MARPOL (International Convention for the Prevention of Pollution from Ships, 1973, 1978, 1997), Annex VI (Prevention of Air Pollution from Ships, enforced since 19 May 2005). The $NO_x$ emission limits for marine diesel engines with a rated power of more than 130 kW are divided into Tiers I, II and III according to the IMO (International Maritime Organisation). The limit values are applied depending on the date of construction date and the area of navigation, as shown in Figure 1.

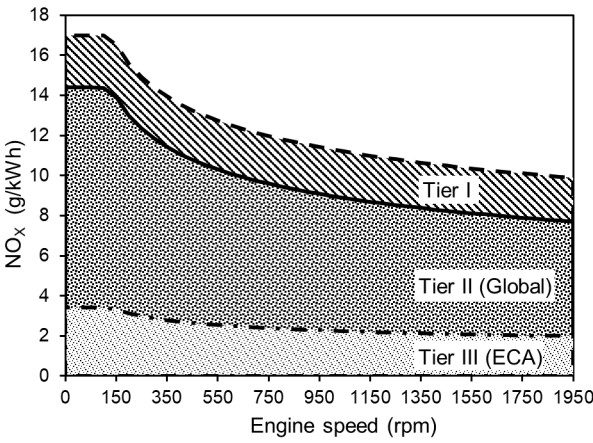

**Figure 1.** $NO_x$ emission limits for marine engines [1].

Tier I refers to all ships built since 2000. Tier II is enforced since 2011. Due to Tier II, the $NO_x$ emission limits are reduced by up to 21% compared to Tier I. Tier III requires an additional 76% reduction in emissions reduction for ships operating in ECA (Emission Control Areas). Depending on their operating area of navigation, many ships are affected by Tiers II and III. It is therefore necessary to optimize the emissions of marine diesel engines.

Most merchant ships are powered by a two-stroke low-speed diesel engine whose overall efficiency exceeds 50% under certain operating conditions. Medium-speed diesel engines are half the size at the same rated power and $NO_x$ emissions are considerably lower. However, their efficiency does not exceed 48%. The advantages of medium-speed diesel engines are particularly pronounced in diesel-electric and hybrid systems. The slightly higher specific fuel consumption of four-stroke diesel engines can be compensated by utilizing waste heat of the exhaust gases and cooling water.

In order to meet the environmental requirements for medium-speed diesel engines, various measures are applied to reduce emissions. These measures are divided into primary and secondary measures. Primary measures involve modifying the process in the engine cylinder. Secondary measures include exhaust after-treatment. Fuel type and quality also have a significant influence on emissions. Technologies for reducing $NO_x$ emissions are listed in Table 1.

**Table 1.** $NO_x$ emission reduction technologies [2].

|  | $NO_x$ **Emission Reduction Technology** | **Expected Reduction** |
|---|---|---|
| 1 | Two-stage turbocharger and Miller process | ~40% |
| 2 | Combustion process adjustment | ~10% |
| 3 | EGR—exhaust gas recirculation | ~60% |
| 4 | Higher humidity of the scavenging air | ~40% |
| 5 | Adding water to the fuel before injecting | ~25% |
| 6 | Direct injection of water into the cylinder | ~50% |
| 7 | SCR—selective catalythic reduction | ~80% |
| 8 | Replacing liquid fuel with gaseous fuel | ~85% |

$NO_x$ emission reduction technologies, which are marked 1, 2, 7 and 8 in Table 1, have the most favourable impact on energy efficiency and specific fuel oil consumption (SFOC). The implementation of other listed technologies leads to an increase in specific fuel consumption.

The adjustment of the combustion process in the engine cylinder by increasing the compression ratio while simultaneously reducing the amount of fuel injected per crankshaft revolution theoretically enables the approximately constant pressure of the combustion process. This leads to lower maximum pressure and a lower maximum temperature, which is beneficial because $NO_x$ emissions

are largely temperature-dependent. By using modern electronically controlled fuel injection systems, this technology does not lead to a significant increase in specific fuel consumption.

The Miller process is specific to the early closing of the inlet valves closing. R. Miller investigated the influence of the closing angle of the inlet valve on the temperature at the end of compression in the cylinder. Early closing of the inlet valve results in additional air cooling in the cylinder due to expansion. Pressure and temperature at the end of compression are lower compared to the process where the inlet valve closes at the later angle. This technology shortens the compression duration and prolongs the expansion, resulting in a reduction in $NO_x$ emissions and a reduction in specific fuel oil consumption. To achieve the same engine power, the engine must be equipped with a turbocharger that supplies the same amount of air to the cylinder. More efficient (two-stage) turbochargers with higher pressure ratios are required.

A single-zone, zero-dimensional model of the four-stroke diesel engine was developed to investigate the effects of early inlet valve closure and increased charge pressure on the engine specific fuel oil consumption and changes in pressure and temperature in the engine cylinder [3].

## 2. Numerical Model of a Four-Stroke Diesel Engine

The zero-dimensional numerical model is based on the laws of conservation of energy and mass and solving the resulting differential equations in the way described by Medica [4] and Mrakovčić [5]. The model has additional features such as variable integration step selection, variable inlet valve closing angle, adjustment of the turbocharger air mass flow, graphic display of the results and much more.

The software was developed in the C programming language. The model was validated using data provided by the engine manufacturer and test drive data obtained from the operation of Wärtsilä W12V50DF engine with a rated power of 11,700 kW while running a synchronous three-phase generator.

The main advantages of the applied model compared to multidimensional and multi-zone models are lower complexity, higher execution speed, adaptability and satisfactory accuracy of the obtained results, which are comparable to more complex models.

The medium-speed diesel engine consists of the following interconnected subsystems: engine cylinder, inlet manifold, exhaust manifold, turbocharger, intercooler, fuel injection subsystem, associated piston mechanism and valve timing mechanism, as seen in Figure 2.

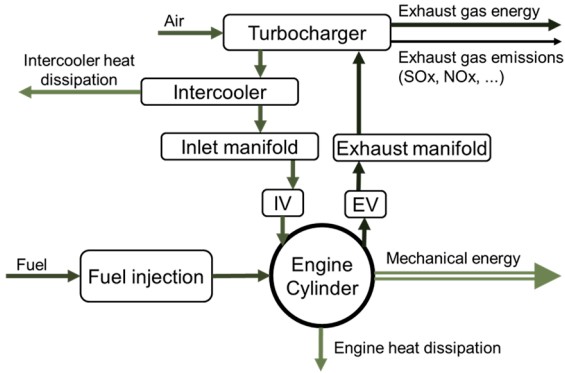

**Figure 2.** Diesel engine subsystems within implemented zero-dimensional numerical model.

The control volumes are interconnected by appropriate connections, which allow the exchange of the working medium. In the cylinder control volume, the heat is exchanged through the walls between the working medium and the cooling water. The heat is also exchanged with the ambient air through the walls of the inlet and outlet manifold. The heat generated by friction in the bearings is taken into account via the mechanical losses mean pressure, while the heat dissipated by radiation is neglected as it does not exceed 1% of the total heat input. Pressure and temperature in the control volumes are determined by solving differential equations derived from the laws of conservation of energy and mass. The properties of the working medium are determined according to Woschni [6] and Jankov [7].

*2.1. Mass Conservation Law*

The mass change dm in the engine cylinder, inlet and exhaust manifold during the angle of rotation of the crankshaft $d\varphi$ is caused by the flow of the working medium through the inlet and exhaust valves, the mass of the injected fuel and the mass loss due to leakage can be expressed as:

$$\frac{dm_i}{d\varphi} = \frac{dm_{i,in}}{d\varphi} + \frac{dm_{i,ex}}{d\varphi} + \frac{dm_{i,f}}{d\varphi} + \frac{dm_{i,leak}}{d\varphi} \tag{1}$$

where $m_{in}$ is the mass of the medium entering the control volume and $m_{ex}$ is the mass of the medium exiting the volume, $m_f$ is the mass of fuel supplied and $m_{leak}$ is the mass of the medium exiting the volume. In a new or properly maintained engine, the leaked mass may be neglected.

*2.2. Energy Conservation Law*

Energy balance of the medium in the control volume is given by:

$$\frac{dQ_i}{d\varphi} = \frac{dQ_{i,f}}{d\varphi} + \frac{dQ_{i,w}}{d\varphi} + h_{in}\frac{dm_{i,in}}{d\varphi} + h_{ex}\frac{dm_{i,ex}}{d\varphi} + h_f\frac{dm_{i,f}}{d\varphi} - p \cdot dV \tag{2}$$

where $dQ_{i,f}$ denotes heat released through fuel combustion and $dQ_{i,w}$ heat exchanged through the walls. The following variables indicate the sensible heat of the medium entering or leaving the control volume, the heat released by fuel combustion and conducted mechanical work. When the medium flows into the control volume, its enthalpy is added to the energy balance of the control volume; when it flows out, the enthalpy is subtracted.

Assuming that the internal energy of the gas depends solely on temperature, the equation of the temperature change is given by

$$\frac{dT_i}{d\varphi} = \frac{1}{m_i\left(\frac{\partial u}{\partial T}\right)_i}\left[-p_i\frac{dV_i}{d\varphi} + \sum_j \frac{dQ_{i,j}}{d\varphi} + \sum_k h_{i,k}\frac{dm_{i,k}}{d\varphi} - u_i\frac{dm_i}{d\varphi} - m_i\left(\frac{\partial u}{\partial \lambda}\right)_i\frac{d\lambda_i}{d\varphi}\right] \tag{3}$$

In previous equations, all variables containing the mass or enthalpy of the fuel refer only to the control volume in which the fuel burns or to the cylinder. The same applies to variables describing a change in volume. When the fuel burns in the cylinder, the chemical energy of the fuel is converted into heat, which increases pressure and temperature. The increased pressure acts on the piston, where the thermal energy is converted into mechanical work.

*2.3. Indicated Work*

Indicated mechanical work is determined by:

$$\frac{dW_c}{d\varphi} = p_c\frac{dV_c}{d\varphi} \tag{4}$$

The pressure $p_c$ in the cylinder is determined using the equation of state for a gas:

$$p_c = \frac{m_c \cdot R_c \cdot T_c}{V_c} \tag{5}$$

Current cylinder volume $V_c$ is derived from the crankshaft geometry:

$$V_c(\varphi) = \frac{V_s}{2}\left[(1 - \cos\varphi) + \frac{1}{\lambda_m}\left(1 - \sqrt{1 - \lambda_m^2 \sin^2\varphi}\right) + \frac{2}{\varepsilon - 1}\right] \tag{6}$$

where $V_S$ is the cylinder swept volume, $\varepsilon$ is compression ratio and $\lambda_m$ denotes the ratio between crank radius and piston stroke.

### 2.4. Heat Exchange

Heat transfer through the cylinder walls can be expressed as:

$$\frac{dQ_{w,c}}{d\varphi} = \sum_i \alpha_c \cdot A_{w,c,i}(T_{w,i} - T_c)\frac{dt}{d\varphi} \tag{7}$$

According to the research of Pfalum and Mollenhauer [8] and Löhner and Döhring [9], there are no significant temperature changes under stationary operating conditions, therefore a mean temperature is assumed. Furthermore, relatively small deviations in the heat transfer coefficient can be neglected, so that the mean heat transfer coefficient can be applied in the calculations. For the calculation of the heat transfer coefficient, an empirical expression according to Hohenberg [10] is used in this paper:

$$\alpha_c = C_1 \cdot V_c^{-0.06} \cdot p_c^{0.8} \cdot T_c^{-0.4} \cdot \left(c_{mps} + C_2\right)^{0.8} \tag{8}$$

where $C_1$ and $C_2$ are the empirical coefficients and $c_{mps}$ is the mean piston speed.

### 2.5. Heat Release

Heywood [11] and Boulchous [12] divided numerical models that describe the complex process of fuel combustion inside the cylinder into zero-dimensional, quasidimensional and multidimensional models. Vibe [13] provided the heat release rate by the following expression:

$$\frac{x_f}{d\varphi} = C(m+1)\left(\frac{\varphi - \varphi_{IS}}{\varphi_{CD}}\right)^m \exp\left(-C\left(\frac{\varphi - \varphi_{IS}}{\varphi_{CD}}\right)\right)^{m+1} \tag{9}$$

where $x_f$ is the relative proportion of fuel burned, $C$ is the constant that depends on the efficiency of fuel combustion. The index IS refers to the crankshaft angle at which ignition starts, while the index CD represents the duration of combustion. The exponent m is determined according to Woschni and Anisits [14]. The change in combustion duration $\Delta\varphi_{CD}$ is determined according to Betz and Woschni [15].

Sitkei [16] provided the expression for determining the ignition delay for diesel fuel and Boy [17] adjusted the expression for heavy fuel. A simpler expression according to Wolfer [18] was implemented in the developed numerical model.

It is assumed that the rate of injected fuel mass follows the heat release rate and that the combustion products are immediately mixed with the medium in the cylinder to form a homogeneous mixture. The total mass within the cylinder increases during combustion due to the injected fuel. The excess air in the engine cylinder is calculated from the mass of the gases in the engine cylinder and the mass of the injected fuel.

### 2.6. Change in Mass and Excess Air in the Cylinder

The change in mass in the engine cylinder due to fuel injection is expressed by:

$$\frac{dm_c}{d\varphi} = \frac{dm_{f,c}}{d\varphi} = \frac{dx_f}{d\varphi}m_{f,proc} = \frac{1}{\eta_{comb}LHV}\frac{dQ_f}{d\varphi} \tag{10}$$

Additionally, fuel injection affects the change in excess air ratio which is calculated as follows:

$$\frac{d\lambda_c}{d\varphi} = -\frac{\lambda_c}{m_{f,c}}\frac{dm_{f,c}}{d\varphi} \tag{11}$$

There is no change in the gas composition and there is no change in the excess air ratio as the working medium flows out of the observed control volume. If gases flowing into the control volume

have a different composition, there is also a change in the excess air ratio. The change in airflow as a function of the crankshaft angle is determined by the expression:

$$\frac{d\lambda_c}{d\varphi} = = \frac{\frac{dm_{c,i}}{d\varphi}\left(1 - \frac{\lambda_c S_{AFR}+1}{\lambda_i S_{AFR}+1}\right)}{S_{AFR} m_{g,c}} \tag{12}$$

where $S_{AFR}$ is the stoichiometric mass of air in mixture with fuel.

### 2.7. Working Medium Exchange in a 4-Stroke Engine Cylinder

The working fluid flows between the cylinder and the inlet and exhaust manifolds. The flow of the working fluid from one control volume to the other is determined by valves timing, the effective flow area and the pressure difference:

$$\frac{dm}{d\varphi} = \alpha_p A_{p,geo} \psi p_1 \sqrt{\frac{2}{R_1 T_1}} \frac{dt}{d\varphi} \tag{13}$$

In the displayed expression, the geometric flow areas $A_{p,geo}$ of the inlet and exhaust valves are determined according to the camshaft cam geometry. The flow coefficient αp is determined according to Chapman [19]. The flow function $\psi$ for the subcritical pressure ratio is determined according to Bošnjaković [20]

$$\psi = \sqrt{\frac{\kappa}{\kappa-1}\left[\left(\frac{p_2}{p_1}\right)^{\frac{2}{\kappa}} - \left(\frac{p_2}{p_1}\right)^{\frac{\kappa+1}{\kappa}}\right]}, \quad \text{if } 1 \geq \frac{p_2}{p_1} \geq \left(\frac{2}{\kappa+1}\right)^{\frac{\kappa}{\kappa+1}} \tag{14}$$

and flow function $\psi$ for supercritical pressure is:

$$\psi = \left(\frac{2}{\kappa+1}\right)^{\frac{1}{\kappa-1}} \sqrt{\frac{\kappa}{\kappa+1}}, \quad \text{if } \frac{p_2}{p_1} < \left(\frac{2}{\kappa+1}\right)^{\frac{\kappa+1}{\kappa}} \tag{15}$$

Subscript 1 refers to the state in the upstream control volume, while subscript 2 refers to the state in the downstream control volume.

### 2.8. Turbocharger

For modelling the operation of a diesel engine under stationary operating conditions, the numerical model of the turbocharger does not require the use of suitable compressor map data. Instead, it is acceptable to assume that the air mass flow is known for a given engine load. Engine manufacturers typically provide inlet manifold pressure and air mass flow for a number of operating regimes in the 50% to 100% engine load range. The exhaust gas mass flow through the turbine is determined by the following expression:

$$\frac{dm_T}{d\varphi} = \alpha_T A_{T,geo} \psi p_{EM} \sqrt{\frac{2}{R_{EM} T_{EM}}} \frac{dt}{d\varphi} \tag{16}$$

where $\alpha_T$ is the flow coefficient, $A_{T,geo}$ denotes the cross-sectional area of the turbine, $\psi$ is the flow function and $p_{EM}$ is exhaust manifold pressure.

The temperature of the exhaust gases after the turbine is calculated according to:

$$T_{AT} = T_{EM} - \frac{\left|\Delta h_{T,is}\right|}{\eta_{T,is} \cdot c_{p,EG}} \tag{17}$$

*2.9. Effective Engine Power*

The indicated engine power is determined by integration of the total work of all cylinders during one duty cycle:

$$P_{\text{ind}} = \frac{n_{\text{M}}}{30\tau} \sum_{i=1}^{z} \int \frac{\mathrm{d}W_{\text{C,i}}}{\mathrm{d}\varphi} \mathrm{d}\varphi \qquad (18)$$

where $z$ denotes the number of cylinders and $n_{\text{M}}$ is crankshaft speed in rpm.

Effective engine power is calculated by the following equation:

$$P_{\text{ef}} = \frac{z n_{\text{M}}}{30\tau} V_{\text{S}} p_{\text{mep}} = P_{\text{ind}} \frac{p_{\text{mep}}}{p_{\text{mip}}} \qquad (19)$$

where $p_{\text{mep}}$ is the mean effective pressure and $p_{\text{mip}}$ is the mean indicated pressure. The mean effective pressure is determined by subtracting the mean pressure of the mechanical losses from the mean indicated pressure. The mean pressure of mechanical losses takes into account losses caused by friction and operation of oil and water pumps. In the developed numerical model, the mean pressure of mechanical losses is calculated using approximate expressions according to Maass [21].

## 3. Numerical Model Validation

The developed numerical model was validated using the engine manufacturer data and measurements acquired during test voyages of liquid natural gas (LNG) ships with diesel-electric propulsion.

*3.1. Referenced Engine Wärtsilä 12V50DF*

Validation was based on the engine W 12V50DF manufactured by Wärtsilä. All data presented in Table 2 refer to engine performance running on heavy fuel oil (HFO).

**Table 2.** Data for the engine Wärtsilä 12V50DF [22].

| Engine Parameter | Value |
|---|---|
| Bore, mm | 500 |
| Stroke, mm | 580 |
| Valves per cylinder (inlet/exhaust) | 2/2 |
| Inlet/outlet valve diameter, mm | 165/160 |
| Number of cylinders and configuration | 12 cylinders, V/45° |
| Maximum continuous engine power, kW | 11,700 |
| Engine speed, rpm | 514 |
| Mean piston speed, m s$^{-1}$ | 9.9 |
| Number of turbochargers | 2 |
| Turbocharger type | ABB TPL71-C |

Table 3 presents manufacturer's data of the referenced engine and Table 4 presents data measured during sea trial of an LNG carrier.

**Table 3.** Manufacturer data for W 12V50 DF engine [22].

| Engine Load | 50% | 75% | 100% |
|---|---|---|---|
| Engine power, kW | 5850 | 8775 | 11,700 |
| Specific fuel oil consumption, g/kWh | 196 | 187 | 189 |
| Exhaust gases temperature, °C | 337 | 336 | 352 |
| Exhaust gases mass flow after turbocharger, kg/s | 13.9 | 18.4 | 23.0 |

**Table 4.** Measured data for W 12V50 DF engine (*sea trial*).

| Engine Load | 40% | 50% | 71% |
|---|---|---|---|
| Engine power, kW | 4680 | 5850 | 8307 |
| Specific fuel oil consumption, g/kWh | 199 | 197 | 190 |
| Maximum cylinder pressure, bar | 70 | 83 | 107 |
| Exhaust gases temperature after turbocharger, °C | 412 | 392 | 359 |

*3.2. Model Validation*

The simulation of engine operation under steady-state conditions was performed for five operating regimes in the range of 40% to 100% of the engine rated power. Model validation was performed by comparing data from Tables 3 and 4 with data obtained by simulating specific fuel consumption, maximum cylinder pressure and exhaust gas temperature. Figure 3 shows closed indicating diagrams for five engine operating regimes obtained by developed engine operation simulation software.

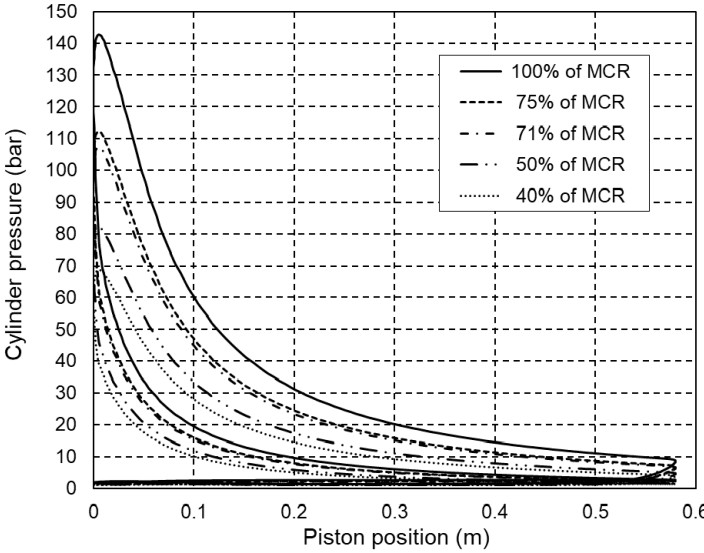

**Figure 3.** Closed indicated pressure diagrams at 40% to 100% of engine rated power.

Figure 4 shows the comparison of the specific fuel oil consumption measured on the testbed and during the sea trial with the values obtained by the engine simulation model. The largest deviation occurs at 40% of the engine load, which is approximately 3.5%, i.e., 7 g/kWh. The smallest deviation from the measured data occurs between 50% and 71% of the maximum engine load and is less than 1%, i.e., 2 g/kWh.

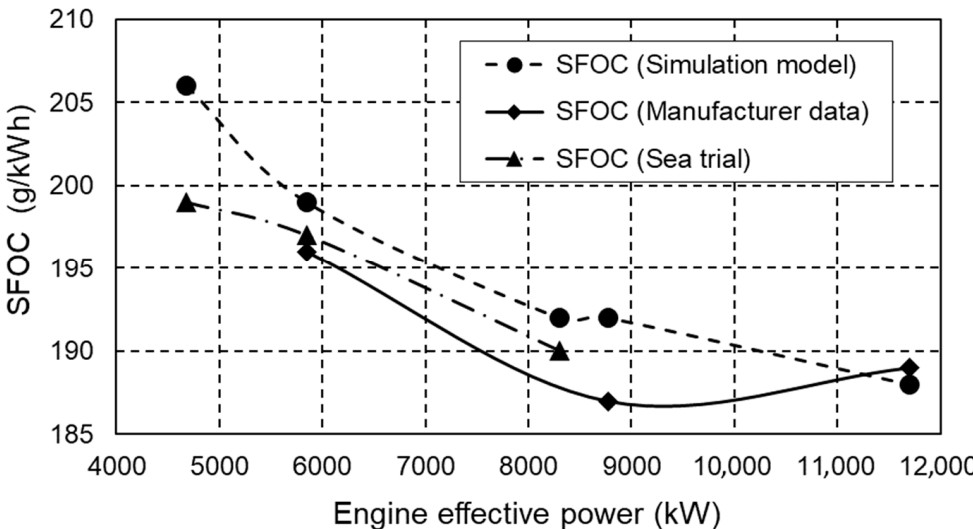

**Figure 4.** Comparison of the specific fuel oil consumption.

Figure 5 shows the comparison of exhaust gas temperatures after the turbocharger. The biggest difference occurs at 50% of the engine load and its value is 10.4%, i.e., 35 °C. The deviation at the same operating point compared to the sea trial data is 6.8%, i.e., 20 °C. The difference between the exhaust gas temperatures after the turbocharger is only 1.7%, i.e., 6 °C at full engine load.

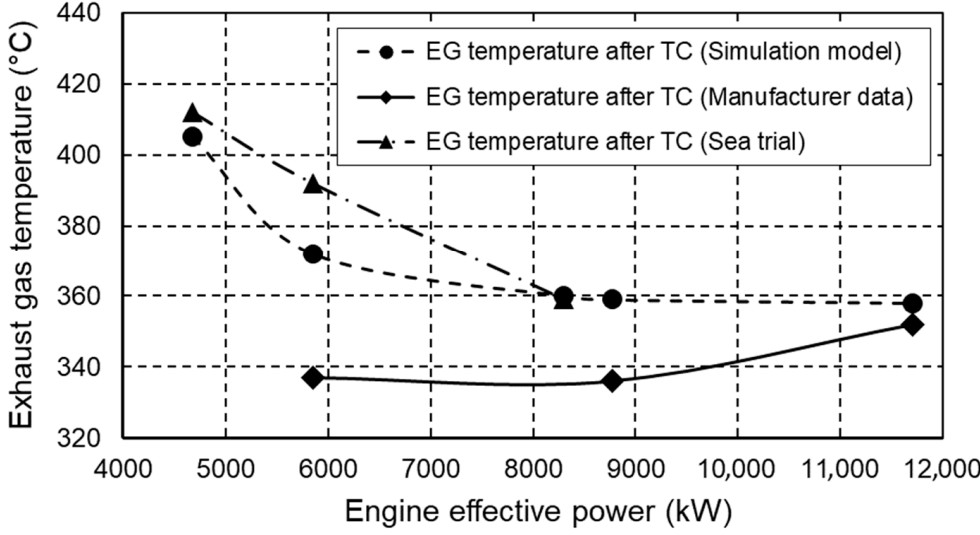

**Figure 5.** Comparison of the exhaust gas temperatures after turbocharger.

Figure 6 shows a comparison of the maximum pressures in the engine cylinder. The absolute pressure deviations are less than 1 bar at all observed operating regimes. The maximum pressure in the engine cylinder measured during the sea trial is presented as an average value of all 12 cylinders.

Figure 7 shows a comparison of exhaust gas mass flows as a function of engine load obtained by the simulation model and the manufacturer's data. A small deviation of the values can be observed from the diagram, which further confirms the applicability of the simulation model.

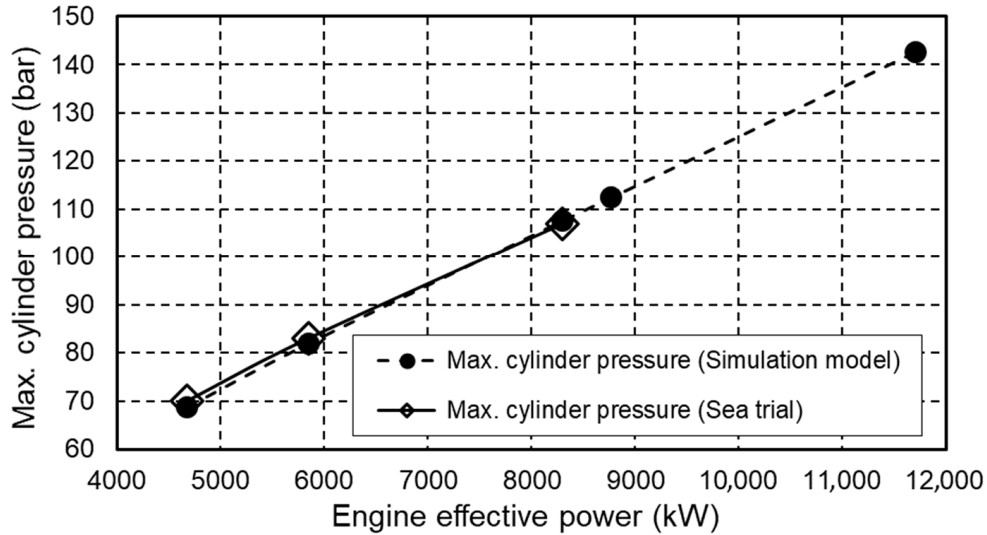

**Figure 6.** Comparison of maximum pressures in the engine cylinder.

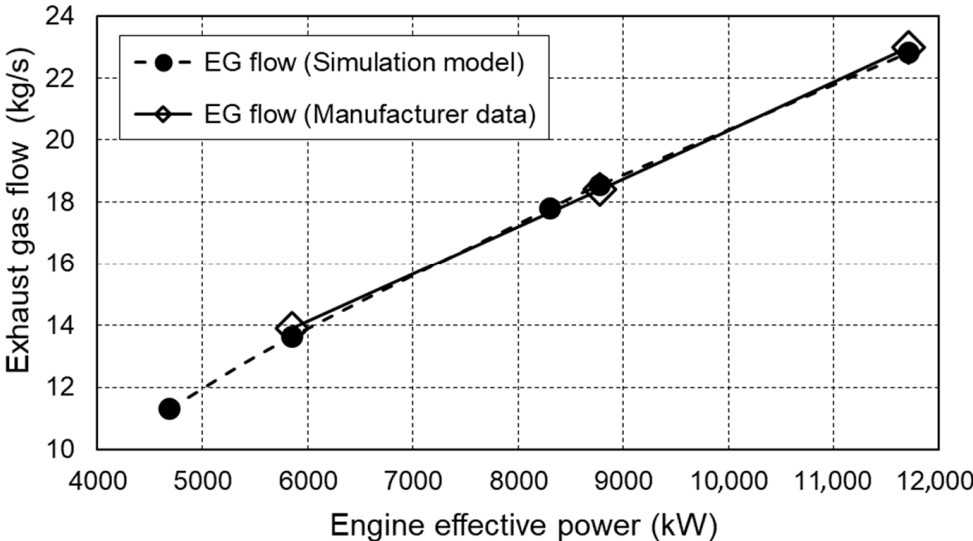

**Figure 7.** Comparison of exhaust gas flow at different engine load.

## 4. Analysis of the Impact of Earlier Inlet Valve Closing Angle

The early closing of the inlet valves in the Miller process (also known as Miller timing) is an effective measure to reduce $NO_x$ emissions and fuel oil consumption in the internal combustion engine. This measure can be applied to almost all engine types. Its positive effects are highlighted by the use of an efficient turbocharger. In the Miller process, an over-expansion process in the engine is achieved by shortening the compression stroke by closing the inlet valve earlier or later compared to conventional valve timing.

Miller [23] mainly intended to increase engine power, but his concept is now used to reduce $NO_x$ emissions and fuel oil consumption. Closing the inlet valve too early or too late with unchanged inlet manifold pressure results in poor cylinder filling which means that pressure and temperature are lower at the end of the compression stroke. Therefore, temperature and pressure in the high-pressure part of the process are also lower. Since the formation of $NO_x$ is exponentially related to temperature, even a small drop in temperature causes a significant decrease in $NO_x$ formation. The possibility of detonation combustion is also reduced.

Miller [24] has pointed out the possibility of applying this concept also to engines running on gas fuel. Clarke and Smith [25] carried out tests on engines with controllable inlet valve timing and analyzed the ability of the Miller process to increase engine power. Wang and Ruxton [26], and Wang [27] emphasize the ability of the Miller process to reduce $NO_x$ emissions. Ust [28] compared the Otto-Miller, Diesel-Miller and Dual Miller processes. Gonca and Sahin [29] analyzed the influence of engine design and operating parameters on the performance of a turbocharged diesel engine running with the Miller process. Guan [30] and Tagai [31] investigated the possibility of combining the Miller process with other $NO_x$ reduction measures. Gonca [32] presented the detailed thermodynamic analysis of the dual Miller cycle for marine diesel engines. The results of the above-mentioned investigations were an additional motive for investigating the effects of early inlet valve closure on the performance of a medium-speed marine diesel engine. In the following, the results of the engine operation simulation using the developed numerical engine model are presented. The results obtained are compared with the values measured on the base engine, as displayed in Table 5.

**Table 5.** Base engine data.

| Engine Parameter | Value |
|---|---|
| Bore, mm | 460 |
| Stroke, mm | 580 |
| Number of inlet/exhaust valves per cylinder | 2/2 |
| Inlet/exhaust valve diameter, mm | 160/157 |
| Inlet valve closing angle (° after BDC) | 26° |
| Number of cylinders and configuration | 6 cylinders, inline |
| Engine speed, rpm | 600 |
| Mean piston speed, m s$^{-1}$ | 11.6 |
| Engine maximum continuous rating, kW | 7200 |
| Number of turbochargers | 1 |

The research was conducted in two phases. In the first phase, the closing angle of the inlet valve was shifted to an earlier angle in three successive steps, whereby minor corrections were made to the inlet manifold pressure. In the second phase, the inlet valves were closed 34° before bottom dead center (BDC), i.e., 60° earlier than the base engine, and the inlet manifold pressure was increased by 10% and 20% respectively.

*4.1. Simulation Results of Engine Performance with Early Inlet Valve Closing*

The closing angle of the inlet valve of the base engine is set to 26° after BDC. Simulations were performed for earlier inlet valve closing angles, at 6° after BDC, 14° before BDC and 34° before BDC, as shown in Figure 8. The mean inlet manifold pressure was increased only as much as necessary to compensate for the decrease in air mass in the engine cylinders due to earlier inlet valve closure. In addition, simulations at 50%, 75% and 100% of the maximum engine power were performed to investigate the behaviour of the engine parameters of interest.

The obtained closed indicating diagram and the change in mean temperature in the engine cylinder at 100% of engine load are shown in Figures 9 and 10. It can be seen that the peak pressure and mean temperature in the engine cylinder decrease and decrease with earlier closing of the inlet valve. Figure 9 shows the mean temperature in the engine cylinder at maximum engine load during the fuel combustion and expansion stroke, so that details of interest can be seen more clearly. The temperature drop is particularly noticeable at closing angles of 40° or 60° earlier than in the base engine, where lowering the mean temperature in the engine cylinder leads to lower $NO_x$ formation. The simulation results for 50% and 75% of the maximum engine load show similar trends.

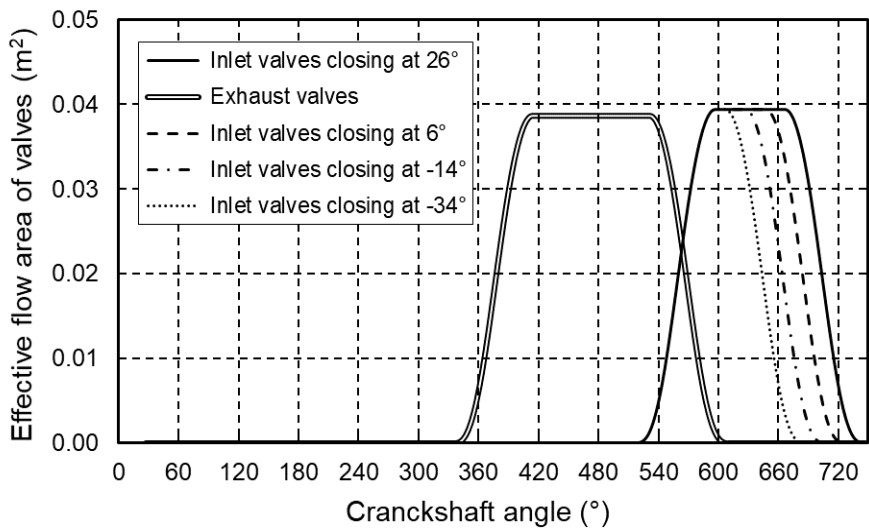

**Figure 8.** Inlet and exhaust valve timing used in simulation.

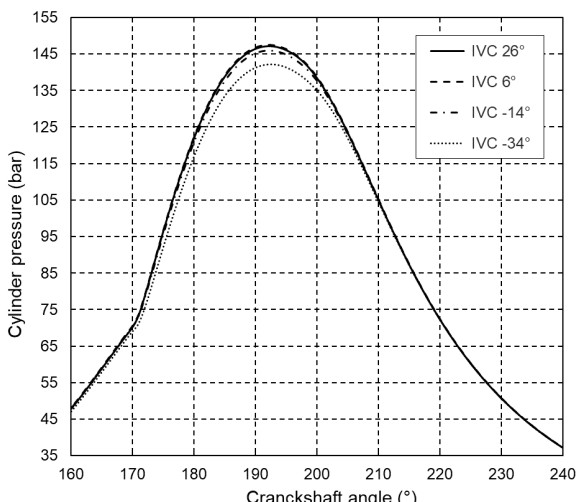

**Figure 9.** High-pressure part of indicated diagram at 100% of maximum engine load.

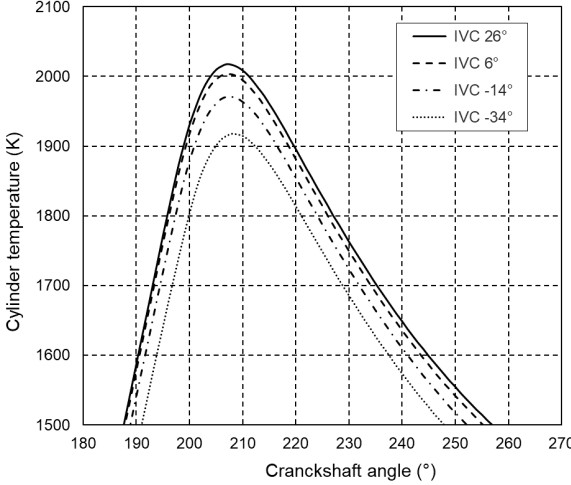

**Figure 10.** Mean temperature in the engine cylinder at 100% of maximum engine load.



The specific fuel oil consumption also decreases as inlet valves close earlier than on the base engine. The simulation results presented in Figure 11 show that the specific fuel oil consumption becomes lower at inlet valve closing angles between 20° and 40° earlier than with the base engine. If the inlet valve closing angles are set to more than 40° before BDC, the specific fuel oil consumption increases. It can also be seen that the specific fuel oil consumption decreases with higher engine load.

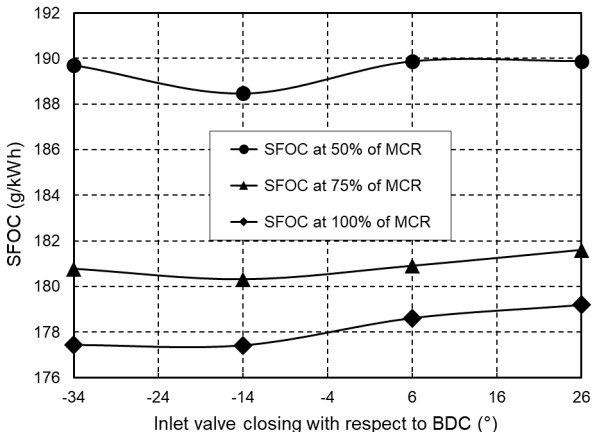

**Figure 11.** Specific fuel oil consumption depending on the inlet valves closing angle.

The change of the mean temperature of the engine cylinder with the inlet valve closing angle for different engine loads is shown in Figure 12. As expected, the mean engine cylinder temperatures increase with higher engine load. To allow a comparison of the results, a correction of the mean pressure in the engine inlet manifold is necessary to allow the engine to develop the required power at different closing angles of the inlet valves. This condition is met by setting a suitable mean pressure in the engine inlet manifold to allow the required mass of air for the complete fuel combustion. The change in the required mean pressure in the engine inlet manifold with an inlet valve closing angle for different engine loads is shown in Figure 12.

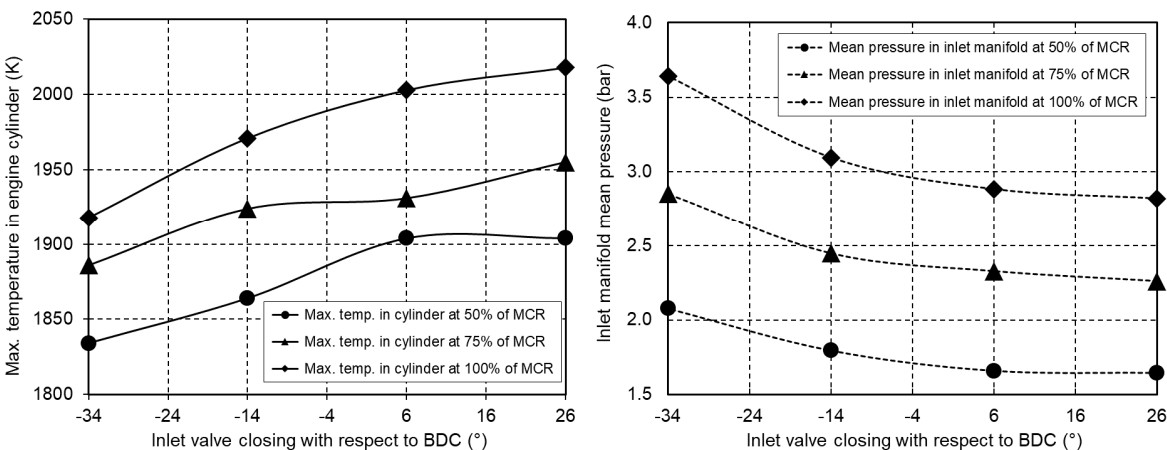

**Figure 12.** Maximum temperature in the engine cylinder and inlet manifold mean pressure depending on the inlet valves closing angle.

All the results obtained confirm the effectiveness of the Miller process in lowering the temperature of the high-pressure part of the process and in reducing $NO_x$ emissions while increasing the efficiency of the process, which is reflected in lower specific fuel oil consumption. It is also noted that the trend in specific fuel oil consumption is at a certain minimum and that the strategy of closing the inlet valves early has its limitations. This research was carried out with the intention of investigating the advantages of early closing of the inlet valves in an engine with an almost unchanged turbocharger

configuration. The results obtained can be used as a solid basis for the simulation of engine performance with increased inlet manifold pressure, which can be achieved by using a highly efficient turbocharger.

### 4.2. Simulation Results of Engine Performance with Increased Inlet Manifold Pressure

In addition to the previously presented simulation of the early closing of the inlet valves, a simulation of the engine performance with increased pressure in the inlet manifold was also carried out. One of the ways in which an increase in inlet manifold pressure can be achieved is by using a highly efficient turbocharger that does not require any significant changes to the engine. This simulation is carried out with inlet valves closed 60° before base engine setup and with inlet manifold pressure increased by 10% and 20% respectively in relation to the base engine. The simulation was performed with 3 different engine loads, i.e., with 50%, 75% and 100% of the maximum engine load.

Figure 13 shows obtained closed indicated diagrams at 100% of maximum engine load with normal inlet manifold pressure (operating point 1) and with pressure increased by 10% and 20% (operating points 2 and 3). As expected, a higher inlet manifold pressure leads to a higher peak pressure during the combustion stroke but the total increase of cylinder peak pressure does not exceed 15%. A diagram of the change in maximum mean temperature in the engine cylinder as a function of inlet manifold pressure is shown in Figure 14. The diagram shows that as the inlet manifold pressure increases, the maximum mean temperature in the engine cylinder decreases due to the higher air-to-fuel ratio.

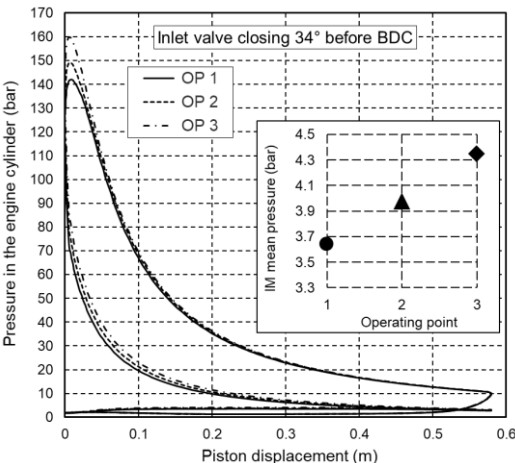

**Figure 13.** Closed indicated diagrams at different inlet manifold pressures at 100% of maximum engine load.

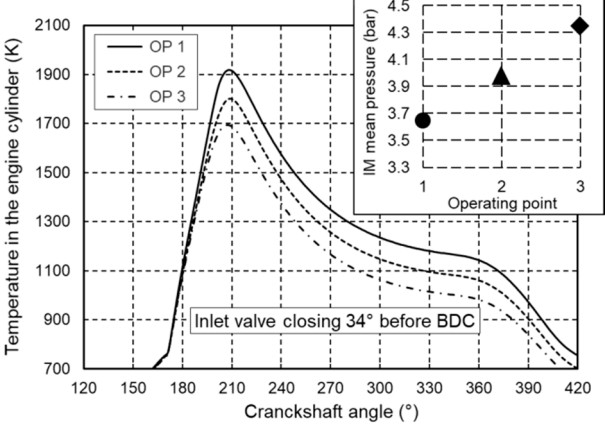

**Figure 14.** Change of mean temperature in engine cylinder at different inlet manifold pressures at 100% of maximum engine load.

Figures 15 and 16 show the influence of the inlet manifold pressure on the specific fuel oil consumption and the maximum mean cylinder temperature. Earlier closing of the inlet valve in combination with an increased inlet manifold pressure leads to a reduction in the specific fuel oil consumption by almost 3% as well as to a decrease in the mean cylinder temperature by about 10%, which also leads to a reduction in $NO_x$.

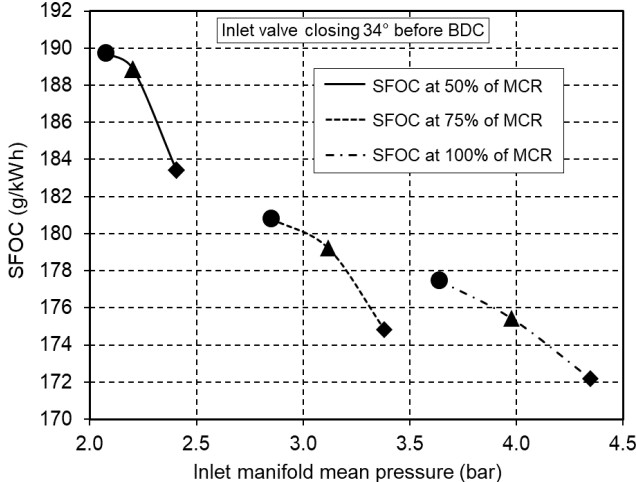

**Figure 15.** Influence of the inlet manifold pressure on specific fuel oil consumption.

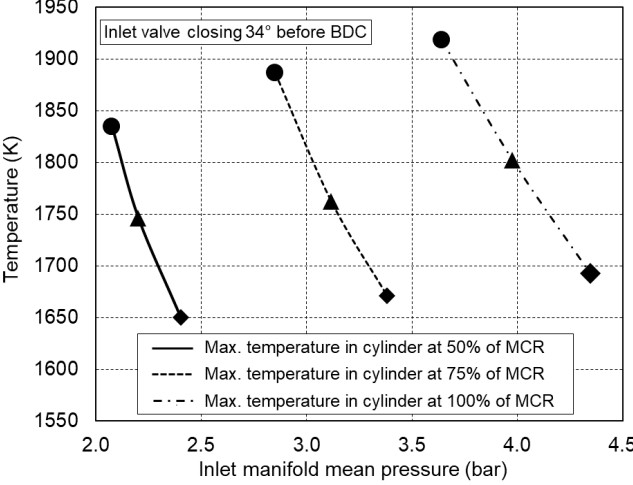

**Figure 16.** Influence of the inlet manifold pressure on maximum engine cylinder temperature.

Figure 17 shows the change in exhaust gas temperature before the turbocharger depending on the closing angle of the inlet valve. In order to achieve the required engine power, the mass of the working medium in the engine cylinder is maintained as with the base engine. It can be seen from the diagram that the earlier closing of the inlet valve lowers the temperature of the exhaust gases. This is caused by the lower pressure and temperature of the working medium in the engine cylinder at the end of the compression.

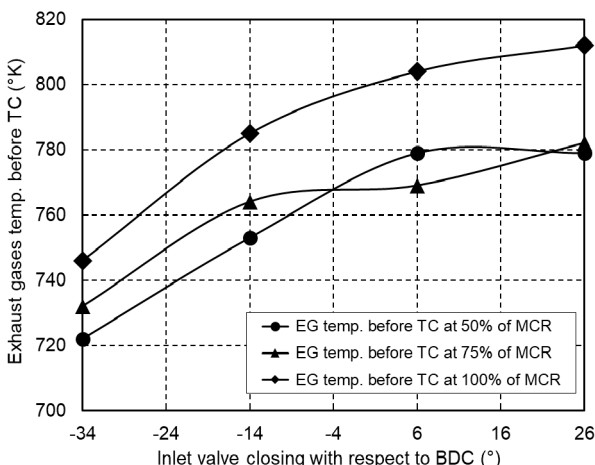

**Figure 17.** Exhaust gas temperature before turbocharger depending on the inlet valves closing angle.

## 5. Conclusions

The results obtained show the efficiency of the Miller process not only in reducing the cylinder temperature, but also in increasing the energy efficiency of the medium-speed marine diesel engine. The increase in efficiency or the reduction in fuel oil consumption is due to the extension of the expansion stroke with a simultaneous increase in the pressure in the inlet manifold.

The analysis of the results obtained shows that the early closing angle of the inlet valve can lower the peak cylinder pressure and the mean engine cylinder temperature, resulting in a reduction in mechanical and thermal stresses, and a reduction in $NO_x$ emissions. For the examined engine, specific fuel oil consumption is lowest when the inlet valves close between 20° and 40° earlier than in the base engine. Although this trend does not continue when the inlet valve closes 60° before the BDC, the specific fuel oil consumption is still lower compared to the base engine. Although it is theoretically possible to operate the engine even with the earlier closing angle of the inlet valves, the cam shape of the camshaft is the practical limitation.

The results obtained by simulation also show that a 20% increase in inlet manifold pressure can reduce specific fuel consumption by almost 3% in engines with early inlet valve closing angle, while the mean cylinder temperature is reduced by about 10%. Lowering the temperature of the high-pressure part of the process results in a reduction in $NO_x$ emissions. At the same time, the increase in peak cylinder pressure does not exceed 15% even at maximum continuous engine power.

The simulation of the operation of the medium-speed marine engine at different loads confirmed the positive effects of the Miller process not only on the reduction in the temperatures of the high-pressure part of the process in the engine cylinder and thus on the formation of $NO_x$, but also on the reduction in specific fuel oil consumption.

The medium-speed marine diesel engines are usually used in the operating mode of about 75% of the nominal continuous output, in which the power plant generators in ships with diesel-electric propulsion are operated. In this operating mode, it is possible to further increase the pressure in the engine inlet manifold by using a suitable turbocharger, which would have additional positive effects on the engine performance due to the earlier closing angle of the inlet valve.

**Author Contributions:** Formal analysis, V.M.-V. and M.V.; Funding acquisition, M.V.; Investigation, V.P. and T.M.; Resources, M.V.; Validation, V.P. and T.M.; Visualization, V.M.-V.; Writing—original draft, V.P.; Writing—review & editing, T.M. and V.M.-V. All authors have read and agreed to the published version of the manuscript.

**Funding:** This work was partially supported by the Croatian Science Foundation under the project IP-2018-01-3739. This work was also supported by the University of Rijeka (project no. uniri-tehnic-18-18 1146 and uniri-tehnic-18-266 6469).

**Conflicts of Interest:** The authors declare no conflict of interest.

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
