# Peer review of "Effect of Early Closing of the Inlet Valve on Fuel Consumption and Temperature in a Medium Speed Marine Diesel Engine Cylinder"

_jmse, doi:10.3390/jmse8100747_

Round 1
Reviewer 1 Report
The paper is devoted to the study of the effects of early inlet valve closure on the performance of a medium-speed marine diesel engine.
The authors had analyzed the main ways of NOx reduction. A well-grounded conclusion has been made about the effect of the temperature in the engine cylinder on the NOx value, and a mathematical model with sufficient accuracy has been developed.
There are several notes on the article:
- The work does not show studies and does not provide data on the effect of early closing of the inlet valve on engine power, which is used in the developed mathematical model (equations 22,23).
- In fig. 12 graph lines do not contain engine load designation
Author Response
Response to Reviewer 1 Comments
Point 1: The work does not show studies and does not provide data on the effect of early closing of the inlet valve on engine power, which is used in the developed mathematical model (equations 22,23).
Response 1:
Since the tested engine is used to drive an electric generator, it must deliver a power corresponding to the current consumption of the electric consumers at all times. The engine delivers less power than the base engine if the intake valves are closed earlier. In order to maintain the same power in the engine with the earlier closing of the intake valves, the mean pressure of the intake manifold had to be increased to compensate for the lack of working medium in the engine cylinder.
In order to be able to compare the results of both engines in the generator mode, the simulation was performed for two engines of equal power. Therefore, the influence of the earlier closing of the intake valves on the engine power has been omitted.
This is mentioned in the manuscript in Chapter 4.1 and in lines 324 and 326 in corrected manuscript.
Point 2: In fig. 12 graph lines do not contain engine load designation
Response 2: The description of fig 12. will be extended in the following way:
"Figure 13. Closed indicated diagrams at different inlet manifold pressures at 100% of maximum engine load".
Due to including an additional Figure in the corrected manuscript (according to other Reviewers request), figure number had to be adjusted from 12 to 13.
Reviewer 2 Report
The single-zone and zero-dimensional model of a four-stroke diesel engine is judged as a paper that analyzes the characteristics of the initial inlet valve closure, the increase in the filling pressure for engine specific fuel oil consumption, and the change in pressure and temperature of the engine cylinder. Please correct and supplement by presenting your opinion as follows. 1. First, it is necessary to clearly present a comparison between the combustion model used in this study and the experimental results. 2. Research on combustion and exhaust characteristics has been conducted through the opening and closing timing of the intake and exhaust using the Miller cycle. In this study, the fuel economy characteristics were analyzed by controlling the opening and closing timing of the intake valve. It is thought that it is necessary to clearly present the results for the parameters of the intake valve. 3. It is also necessary to optimize the characteristics of the turbocharger for this adjustment. It is necessary to present the results for this. 4. In this study, research results were presented on combustion pressure, SFOC, and temperature. It is also necessary to present the analysis results of exhaust gas through the opening and closing timing of the intake valve suggested in this study.Author Response
Response to Reviewer 2 Comments
Point 1: First, it is necessary to clearly present a comparison between the combustion model used in this study and the experimental results.
Response 1: The combustion model used in this paper is based on the Vibe fuel combustion heat release model. The fuel combustion rate and the consequent heat release are described by an exponential function whose parameters are selected based on a test similar to the engine for which the simulation is performed. Such a model can accurately predict the heat release and temperature change in the engine cylinder. These data are guidelines for assessing the process in the engine cylinder and estimating NOx emissions.
The experimental results used in this paper are the results published by the engine manufacturer. These results were used to calibrate the numerical engine model¸ and to check the achieved maximum pressure and temperature during the combustion process. A different comparison of experimental results with the combustion model is difficult to perform due to a different type of data and their characteristics.
Point 2: Research on combustion and exhaust characteristics has been conducted through the opening and closing timing of the intake and exhaust using the Miller cycle. In this study, the fuel economy characteristics were analyzed by controlling the opening and closing timing of the intake valve. It is thought that it is necessary to clearly present the results for the parameters of the intake valve.
Response 2: Research on engine characteristics using the Miller cycle has been conducted only through the opening and closing timing of the intake valve, while exhaust valve timing remained unchanged in regards to the base engine. The simulations have been performed for four closing angles of the inlet valve as described in paragraph 4.1. The closing angle of the inlet valve of the base engine is set to 26° after BDC, while earlier inlet valve closing angles are set at 6° after BDC, 14° before BDC and 34° before BDC. Inlet valve diameter is specified in Table 5, while Figure 8. shows the change of effective flow areas of inlet and exhaust valves in regards to the crankshaft angle.
Point 3: It is also necessary to optimize the characteristics of the turbocharger for this adjustment. It is necessary to present the results for this.
Response 3: The use of the Miller process and the earlier closing of the intake valves have the effect of reducing the mass of the working medium that fills the engine cylinders. In order to maintain the power of the basic engine, it is necessary to create conditions so that the appropriate mass of air can be brought into the engine cylinder. To ensure this, the turbocharger must achieve a higher pressure in the engine intake manifold. Therefore, a calculation was made of how much mean pressure is required in the intake manifold to deliver the required power when the Miller process is applied. This is described in the manuscript in lines 352 to 359. The change in the required mean pressure in the intake manifold with the intake valve closing angle for different engine loads is shown in Figure 12. Characteristics of a turbocharger at the earlier closure of intake valves can be determined from these data.
Point 4: In this study, research results were presented on combustion pressure, SFOC, and temperature. It is also necessary to present the analysis results of exhaust gas through the opening and closing timing of the intake valve suggested in this study.
Response 4: Authors will provide additional diagram which show analysis results of exhaust gas. Figure 17 shows the change of the exhaust gas temperature before turbocharger in regard to inlet valve closing angle for 50%, 75% and 100% of engine rated power.
Round 2
Reviewer 2 Report
Overall, the points pointed out were well reflected. So I accept this paper.